# High Removal Efficiency of Diatomite-Based X Zeolite for Cu^2+^ and Zn^2+^

**DOI:** 10.3390/ma14216525

**Published:** 2021-10-29

**Authors:** Guangyuan Yao, Yuqiang Liu, Shuilin Zheng, Ya Xu

**Affiliations:** 1State Key Laboratory of Environmental Criteria and Risk Assessment, Chinese Research Academy of Environmental Sciences, Beijing 100012, China; slxw.yao1990@hotmail.com; 2Research Institute of Soil and Solid Waste, Chinese Research Academy of Environmental Sciences, Beijing 100012, China; liuyq@craes.org.cn; 3School of Chemical and Environmental Engineering, China University of Mining and Technology (Beijing), Beijing 100083, China; shuilinzheng8@gmail.com

**Keywords:** X zeolite, Cu^2+^ and Zn^2+^, adsorption thermodynamic, adsorption kinetics

## Abstract

Diatomite-based X zeolite was obtained and its crystallinity, morphology, and interface properties were investigated by XRD, BET, SEM, EDS, and XRF. The obtained X zeolite possessed a unique meso-microporous structure and showed good ion exchange properties for Cu^2+^ and Zn^2+^. The pseudo-second-order model and Langmuir isotherm model can best describe the adsorption kinetics and isotherms of Cu^2+^ and Zn^2+^, respectively. The maximal adsorption capacities of X zeolite for Cu^2+^ and Zn^2+^ were 146 and 195 mg/g at 323 K, respectively. Meanwhile, the adsorption process for Cu^2+^ and Zn^2+^ were chemical adsorption and ion exchange, respectively. Furthermore, the adsorption data turned out to be an endothermic and spontaneous process. Compared with other reported materials, the adsorption capacity of X zeolite synthesized from diatomite was among the highest. Therefore, it could be a promising adsorbent for the disposal of wastewater that contains metal ions.

## 1. Introduction

Nowadays, Water pollution has turned out to be more and more serious in the world and especially in developing countries. Heavy metal pollution [1] usually occurred from many industries, such as electroplating, metal mining operations, battery manufacturing, tanneries, fertilizer industries, and leather industries. Furthermore, the US Environmental Protection Agency designated heavy metals as “priority pollutants”, because they are unbiodegradable and tend to accumulate in living organisms [2]. Among these heavy metals, Cu^2+^ and Zn^2+^ are special, because they played an important role in living organisms and act as micronutrients with trace amounts. However, intake of excess Cu^2+^ might cause vomiting, shock, liver and kidney failure, or even death. Excessive intake of Zn^2+^ can also cause serious health problems, such as diarrhea, fever, or even acute gastrointestinal disturbances accompanied by nausea. Therefore, we must regulate the maximum allowed concentrations of Cu^2+^ and Zn^2+^ before releasing them into surface waters. To meet the strict environmental regulations and the rising demands for water quality, the safe disposal of heavy metal pollution has become more and more important and urgent.

Currently, the disposal methods of heavy metals from the aqueous system are usually chemical precipitation [3], coagulation [4,5,6,7], membrane separation [8], photocatalytic degradation [9], extraction [10], chemical oxidation, and reduction [11], biological treatment [12] and adsorption technologies [13,14]. Among all these methods, adsorption is highly effective and economical. In addition, the great advantages of adsorption over others are the low generation of residues, the possibility of large-scale processing, easy metal recovery, and the possibility of the reuse of the adsorbents [15]. Therefore, different types of materials have been developed as adsorbents for the removal of heavy metals, such as modified natural minerals [16,17], activated carbon [18], polymer [19,20], modified metal oxides [21,22,23] and zeolite [24,25].

Zeolites are well-known adsorbents as they can easily absorb and exchange metal ions with positive charges in their framework because they possess the net negative charges generated from the isomorphic substitution of Si^4+^ by Al^3+^ due to the compensation by alkali and alkaline-earth metal ions. Meanwhile, the greatest advantage of zeolites over other adsorbents is that they also show ion exchange selectivity. Because they have a three-dimensional framework constructed by tetrahedral units and then linked by shared oxygen atoms to create cavities and channels of different dimensions to control ion selectivity in zeolite structures. In addition, they are highly porous aluminosilicates with different cavity structures that offer large internal and external surface areas, which are beneficial for adsorption [26,27].

Here, we synthesized X zeolite from diatomite. And systematically characterized their physicochemical properties via various techniques. Then we evaluated the adsorption performance of the obtained sample via the removal of Cu^2+^ and Zn^2+^. To the best of our knowledge, the removal of Cu^2+^ and Zn^2+^ by X zeolite synthesized from diatomite has not been investigated. This work demonstrates the potential performance of X zeolite for the removal of Cu^2+^ and Zn^2+^ in industrial effluent. Moreover, we systematically studied the adsorption kinetics, isotherms, thermodynamics, and the adsorption mechanisms of X zeolite for Cu^2+^ and Zn^2+^.

## 2. Experiments

### 2.1. Materials

Diatomite (Linjiang, China), copper(II) chloride dehydrate (CuCl_2_·2H_2_O), sodium hydroxide (NaOH), zinc sulfate heptahydrate (ZnSO_4_·7H_2_O), aluminum hydroxide (Al(OH)_3_), hydrochloric acid (HCl). The whole reagents were analytical reagent grade, and distilled water was used in all the experiments.

### 2.2. Batch Adsorption Experiments

We synthesized X zeolite as reported in our previous work [26,28]. And then we investigated its adsorption behaviors via bath experiments. First, forty milligrams of X zeolite was added into 40 mL of Cu^2+^ or Zn^2+^ solution of different initial concentrations and different temperatures. Then the concentration of Cu^2+^ or Zn^2+^ solution was measured at adsorption equilibrium and the amount of the adsorbed Cu^2+^ and Zn^2+^ was determined as follows.
(1)$$q_{e}=\frac{v\left(c_{i}-c_{e}\right)}{m\times 1000}$$
where *q_e_* (mg/g) is the adsorption capacity at equilibrium, *v* (mL) is the volume of aqueous solution, *m* (g) is the dosage of the adsorbent, *c_i_* (mg/L) is the initial concentration of Cu^2+^ or Zn^2+^ solution, *c_e_* (mg/L) is the equilibrium concentration of Cu^2+^ or Zn^2+^ solution.

Similarly, the effects of initial solution pH on the adsorption of Cu^2+^ or Zn^2+^ were studied by adjusting the pH value. First, we studied the adsorption kinetics and added 40 mg of X zeolite into 40 mL of Cu^2+^ or Zn^2+^ solutions with an initial concentration of 200 mg/L, 250 mg/L, 300 mg/L, and 350 mg/L in order to determine the minimum time needed for equilibrium. Furthermore, we measured the concentrations of Cu^2+^ or Zn^2+^solution at fixed intervals. Then we studied the thermodynamic properties and added 40 mg of X zeolite into 40 mL of Cu^2+^ or Zn^2+^ solution with different initial concentrations at temperatures of 303, 313, and 323 K. All adsorption experiments here were carried out using three replicates and the average values were displayed.

## 3. Results and Discussion

### 3.1. Adsorption Experiments

#### 3.1.1. Kinetic Models

First, we studied the adsorption behavior of X zeolite for Cu^2+^ and Zn^2+^ and then got the relationship between time and adsorption capacity (Figure 1). The adsorption capacity of X zeolite for Cu^2+^ and Zn^2+^ grew fast within 15 min, and then slowly reached up to adsorption equilibrium within 2 h. The easily accessible sorption sites and high surface area of the adsorbent contributed to the rapid adsorption. Meanwhile, the equilibrium adsorption capacities of X zeolite for Cu^2+^ or Zn^2+^ went up at higher concentrations. The equilibrium adsorption capacities for Cu^2+^ and Zn^2+^ increased from 104 and 122 mg/g to 126 and 147 mg/g when Cu^2+^ and Zn^2+^ concentration increased from 200 to 350 mg/L, respectively. The adsorption capacities of Cu^2+^ and Zn^2+^ increased with their initial concentration raising, due to the higher driving force produced by the concentration gradient [29].

Then we utilized the pseudo-first and pseudo-second-order model to evaluate the adsorption mechanism of X zeolite for Cu^2+^ and Zn^2+^ during the adsorption process. The pseudo-first-order model was shown in the following form [30]:ln(q_e_ − q_t_) = lnq_e_ − k_1_t(2)
where k_1_ (min^−1^) is the rate constant of pseudo-first-order reaction, q_e_ (mg/g) is the equilibrium adsorption capacity of X zeolite for Cu^2+^ or Zn^2+^, q_t_ (mg/g) is the adsorption capacity of X zeolite for Cu^2+^ or Zn^2+^ at any time *t*. From the linear plot of ln(q_e_ − q_t_) versus *t* (Figure 2), both q_e_ and k_1_ can be obtained.

The pseudo−second order model was displayed in the following format [31]:(3)$$\frac{t}{q_{t}}=\frac{1}{k_{2}q_{e}^{2}}+\frac{t}{q_{e}}$$
h = k_2_q_e_^2^(4)
where k_2_ (min^−1^) is the rate constant of pseudo-second-order reaction and h [mg/(g·min)] is the initial sorption rate. The rate constants can be determined via the linear plot t/q_t_ against t (Figure 3).

Table 1 and Table 2 summarized the calculated parameters of the pseudo-first and pseudo-second-order model. The correlation coefficients (R^2^) of the pseudo-second-order model ranged from 0.9995 to 0.9997, and the pseudo-second-order model ranged from 0.7519 to 0.0.9867. The correlation coefficients suggested that the adsorption kinetics can best fit the pseudo-second-order model, which indicates that chemical sorption played an important effect during its adsorption process. It may be owing to the exchange of electrons between X zeolite and Cu^2+^ or Zn^2+^ metal ions [32]. Meanwhile, it found that the values of initial sorption rate (h) increased at a higher initial concentration, which indicated that a higher driving force produced by the concentration gradient can promote the adsorption process [22]. Furthermore, the maximum adsorption capacities for Cu^2+^ and Zn^2+^ could reach 126 and 148 mg/g, respectively.

#### 3.1.2. Isotherm Models

To get a deep understanding of the surface property of X zeolite and maximum adsorption capacity, three classic physical isotherm models [32] were utilized to describe the adsorption mechanism of Cu^2+^ or Zn^2+^ by X zeolite.

We used the Langmuir isotherm model to describe the relationship between Cu^2+^ or Zn^2+^ solution and the surface of X zeolite. And the Langmuir isotherm model was expressed in the following format [33]:(5)$$\frac{C_{e}}{q_{e}}=\frac{1}{q_{m}}C_{e}+\frac{1}{q_{m}K_{L}}$$
where K_L_ (L/mg) is the Langmuir constant related to the adsorption energy, q_m_ (mg/g) is the maximum adsorption capacity of X zeolite. From the linear plot of C_e_/q_e_ against C_e_ (Figure 4), Both q_m_ and K_L_ can be determined.

Meanwhile, We used the Freundlich isotherm model to describe the heterogeneous systems [31]. The equation is described in the following format:(6)$$lnq_{e}=\frac{1}{n}lnC_{e}+lnK_{F}$$
where K_F_ is related to the adsorption capacity, n is related to the adsorption intensity. From data correlation (Figure 5), both n and K_F_ can be determined.

The Dubinin-Radushkevich (D−R) isotherm model is generally used to estimate the adsorption mechanism (physical or chemical adsorption) of microporous adsorbents, which was based on the adsorption potential theory. The adsorption potential theory reflected the change in the Gibbs free energy of an adsorbent after adsorbing a unit molar mass of adsorbate. The equation is expressed as follows:(7)$${\mathrm{lnq}}_{e}=\ln q_{m}-\beta \zeta^{2}$$
(8)$$\zeta =\mathrm{RTln}(1+\frac{1}{C_{e}})$$
(9)$$E=\frac{1}{{\left(2\beta \right)}^{\frac{1}{2}}}$$
where q_e_ (mol/g) is the quantity of adsorbate adsorbed by the adsorbent, q_max_ (mol/g) is the quantity of maximum adsorption capacity of the adsorbent (mol/g). R (8.314/(mol·K)) is the universal gas constant, T (K) is the absolute solution temperature. β (mol^2^/J^2^) and ζ (J/mol) are the constants related to adsorption energy and Polanyi adsorption potential, respectively. C_e_ is the equilibrium concentration (mol/L). From the linear plot of lnq_e_ against ζ^2^ (Figure 6), both q_m_ and β can be determined. E is the mean free energy of adsorption, which is applied to distinguish the adsorption type. When the magnitude of E is below 8 KJ/mol, the adsorption process is physical adsorption. When the magnitude of E is between 8 and 16 KJ/mol, the adsorption process is ion exchange. When the magnitude of *E* is above 16 KJ/mol, the adsorption process is chemical adsorption.

We handled the experiments at 303 K, 313 K, and 323 K, and the results are displayed in Figure 7. It can be found that the adsorption capacities of X zeolite for Cu^2+^ and Zn^2+^ reached as the temperature went up, which indicated the endothermic process.

Table 3, Table 4 and Table 5 summarized the calculated parameters of Langmuir, Freundlich, and D-R isotherms. All the correlation coefficient values (R^2^) of the Langmuir, Freundlich, and D-R models are high. However, the Langmuir model correlation coefficient (R^2^) values of Cu^2+^ and Zn^2+^ were a little higher demonstrating that the adsorption data followed well with the Langmuir isotherm. These results indicated that Cu^2+^ and Zn^2+^ were unevenly adsorbed in the form of monolayer coverage. The K_L_ constant of the Langmuir parameters demonstrated the binding affinity between X zeolite and Cu^2+^ or Zn^2+^ [34]. The K_L_ values of Cu^2+^ range from 0.0521 to 0.0865, and the K_L_ values range from 0.0230 to 0.0293. The K_L_ values suggested that X zeolite possesses stronger adsorption of Cu^2+^ than those of Zn^2+^. The *n* values of Freundlich isotherms for Cu^2+^ and Zn^2+^ were all more than 1 at 303, 313, 323 K demonstrating the facile adsorption between X zeolite and Cu^2+^ or Zn^2+^ [35]. Meanwhile, higher adsorption capacities were obtained at higher temperatures suggesting the endothermic nature. In addition, the theoretical maximum adsorption capacities for Cu^2+^ and Zn^2+^ by the D-R isotherm model are higher. It may be due to that there are still many active adsorption sites in the micropore when the surface adsorption of X zeolite reaches saturation. And the internal diffusion rate of heavy metals is slow, which decreased the binding chance between heavy metals and adsorption sites in X zeolites, thus resulting in a smaller adsorption capacity. The values of E for Cu^2+^ are above 16 kJ/mol suggesting that the adsorption process is chemical adsorption. The values of E for Zn^2+^ are between 8 and 16 kJ/mol demonstrating that the adsorption process is ion exchange. The values of E for Cu^2+^ range from 20.27 to 27.32 kJ/mol, and the values of E for Zn^2+^ range from 14.58 to 15.13 kJ/mol. The values of E suggested that X zeolite possesses stronger adsorption of Cu^2+^ than those of Zn^2+^, which is consistent with the characteristics of Cu^2+^ and Zn^2+^. Therefore, the prepared X zeolite probably possessed selective adsorption towards Cu^2+^ [27]. Moreover, the maximum adsorption capacities of X zeolite for Cu^2+^ and Zn^2+^ can reach 146 and 195 mg/g, respectively.

In addition, we compared the adsorption capacities of X zeolite for Cu^2+^ and Zn^2+^ with some other reported materials. As shown in Table 6 and Table 7, the prepared X zeolite possessed higher adsorption capacities for Cu^2+^ and Zn^2+^ than those of other reported adsorbents. Meanwhile, the present synthetic method of X zeolite is environmentally friendly and perhaps low-cost compared with other methods. Therefore, X zeolite prepared from diatomite could be a candidate for removing Cu^2+^ and Zn^2+^ from wastewater.

#### 3.1.3. Adsorption thermodynamics

Generally, we can get more detailed data of internal energy within the adsorption process through the study of thermodynamics. And the thermodynamic parameters for the adsorption of Cu^2+^ and Zn^2+^ can be described in the following format [41].
(10)$${\Delta G}^{0}={\Delta H}^{0}-T{\Delta S}^{0}$$
(11)$${\mathrm{InK}}_{d}=\frac{{\Delta S}^{0}}{R}-\frac{{\Delta H}^{0}}{\mathrm{RT}}$$
(12)$$K_{d}=\frac{q_{e}}{C_{e}}$$
where ΔG^0^ (kJ/mol) is the standard free energy, ΔS^0^ (J/K) is the standard entropy, ΔH^0^ (kJ/mol) is the standard enthalpy, R (8.314/(molK)) is the universal gas constant, T (K) is the absolute solution temperature, K_d_ is the distribution coefficient. Finally, we can get the ΔG^0^, ΔH^0^, and ΔS^0^, and they are presented in Figure 8.

The calculated thermodynamic parameters of Cu^2+^ and Zn^2+^ were listed in Table 8 and Table 9, respectively. The values of ΔH^0^ for Cu^2+^ and Zn^2+^ range from 5.88 to 12.52 KJ/mol, and the values of ΔS^0^ for Cu^2+^ and Zn^2+^ range from 68.92 to 100.92 J/mol. They demonstrated the endothermic nature and good affinity of X zeolite for Cu^2+^ and Zn^2+^ metal ions, respectively. And positive ΔS^0^ values of Cu^2+^ and Zn^2+^ suggested the increased randomness between the interface of X zeolite and Cu^2+^ or Zn^2+^. However, ΔS^0^ for Cu^2+^ and Zn^2+^ went down as their initial concentration went up, which is consistent with the previous literature [42]. The values of ΔG^0^ for Cu^2+^ and Zn^2+^ at three temperatures are all negative indicating the feasible and spontaneous nature. Moreover, ΔG^0^ for Cu^2+^ and Zn^2+^ all went down with temperature went up indicating the spontaneous nature with high affinity [43]. The ΔG^0^ for Cu^2+^ and Zn^2+^ went up as their initial concentration increased, which is also in accordance with the previous work [32].

#### 3.1.4. Effect of Initial pH

Because the solution pH is one of the critical factors that directly influence the solubility of metals, the effect of initial pH on the adsorption capacity of X zeolite was studied in the pH range of 1.0–6.0. Generally, the increase in pH results in a decrease in solubility of metals due to the formation of metal hydroxides of low solubility. Therefore, making metals soluble in solution is critical for adsorption. As shown in Figure 9, the adsorption capacity of X zeolite increased as the pH value increased. According to the experimental data, the initial pH values of Cu(II) and Zn(II) solutions were 4.88 and 5.48, respectively with the concentration of 250 ppm. Some blue and white flocs appeared with increasing the pH value of Cu(II) and Zn(II) solution due to hydrolysis and the higher adsorption capacity at high pH value was due to the hydroxide precipitation [44]. However, the lower adsorption capacity at lower solution pH was owing to the excessive H^+^, which will compete with the heavy metals for adsorption sites.

### 3.2. Characterization of X Zeolite

#### 3.2.1. XRD

XRD patterns of X zeolite and samples after adsorption with Cu^2+^ and Zn^2+^ are shown in Figure 10. After adsorption of Cu^2+^ and Zn^2+^, we can still observe the mineral structure of X zeolite. and the intensity of X zeolite at 2θ = 6.10°, 9.97°, 15.39°, 23.24°, 26.58°, and 30.86° decreased in intensity and samples after adsorption with Cu^2+^ appears to show some amorphous background possibly due to Cu hydroxide precipitation. Moreover, the software Visual MINTEQ was applied to get the state distribution of Cu^2+^ and Zn^2+^ as pH values range from 1–11. When the pH of the Cu^2+^ solution is lower than 5.0, the copper existed in the Cu^2+^ state (Figure 11a). The pH of the Cu^2+^ solution became 5.28 after adsorption with Cu^2+^. When the pH of the Zn^2+^ solution is lower than 7.5, the zinc existed in the Zn^2+^ state (Figure 11b). The pH of the Zn^2+^ solution became 6.25 after adsorption with Zn^2+^. Based on the above analysis, the state of copper and zinc after adsorption of Cu^2+^ and Zn^2+^ are Zn^2+^, Cu^2+^, and a small amount of Cu hydroxide precipitation, respectively. The pH analysis is in agreement with the XRD data.

#### 3.2.2. SEM

The morphology of X zeolite and samples after adsorption with Cu^2+^ and Zn^2+^ are shown in Figure 12 and Figure 13, respectively. The SEM images of X zeolite are an aggregation of small particles in spherical shape according to our previous work [45]. The morphology of X zeolite was almost unchanged with Cu^2+^ and Zn^2+^ dispersed well on X zeolite after adsorption of Cu^2+^ and Zn^2+^, respectively. The compositions of X zeolite and samples after adsorption of Cu^2+^ or Zn^2+^ are presented in Table 10. After adsorption, the composition of sodium greatly decreased. These chemical analyses suggest that Na ions were exchanged with Cu^2+^ or Zn^2+^ by the ion-exchange process.

#### 3.2.3. BET

According to our previous work [27], the obtained X zeolite from diatomite possessed a unique meso-microporous structure with a high BET surface of 453 m^2^/g, which can improve the mass diffusion and transport of Cu^2+^ and Zn^2+^ by utilizing its large cavity volumes and mesopore channels. And then it will increase the effective meso-microporous contact area, which is beneficial for the adsorption process [46]. The adsorption mechanism of X zeolite for Cu^2+^ and Zn^2+^ can be described as follows: first, metal ions diffused to the surface of X zeolite through its liquid membrane; then metal ions diffused from the surface to the interior of X zeolite; finally, metal ions exchanged with cations at the active sites in the X zeolite.

## 4. Conclusions

The X zeolite was synthesized from diatomite displayed a unique meso-microporous structure. The Langmuir isotherm model can best describe the isotherms of Cu^2+^ and Zn^2+^ with the maximum adsorption capacities of 146 and 195 mg/g, respectively. The pseudo-second-order model can best describe the kinetic of Cu^2+^ and Zn^2+^. The adsorption process for Cu^2+^ and Zn^2+^ is chemical adsorption and ion exchange, respectively. Therefore, the prepared X zeolite probably possessed selective adsorption towards Cu^2+^. Moreover, the Langmuir model can best describe the isotherm of Cu^2+^ and Zn^2+^ indicating that Cu^2+^ and Zn^2+^ were unevenly adsorbed in the form of monolayer coverage. In addition, the adsorption processes of Cu^2+^ and Zn^2+^ are both endothermic and spontaneous. The excellent adsorption performance of X zeolite could be attributed to the meso-microporous structure, which improved its mass diffusion and transport of Cu^2+^ and Zn^2+^ by its large cavity volumes and mesopore channels with effective meso-microporous contact area. Therefore, the X zeolite is a promising and potentially cost-effective adsorbent for the efficient disposal of wastewater that contains more metal ions.

## Figures and Tables

**Figure 1 materials-14-06525-f001:**
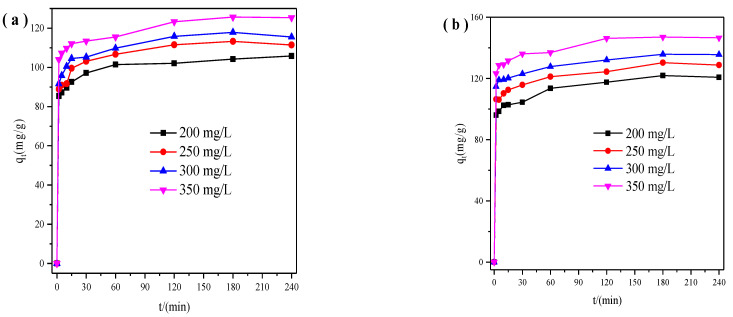
Relationship between time and adsorption capacity of X zeolite for Cu^2+^ (**a**) and Zn^2+^ (**b**) with different concentrations.

**Figure 2 materials-14-06525-f002:**
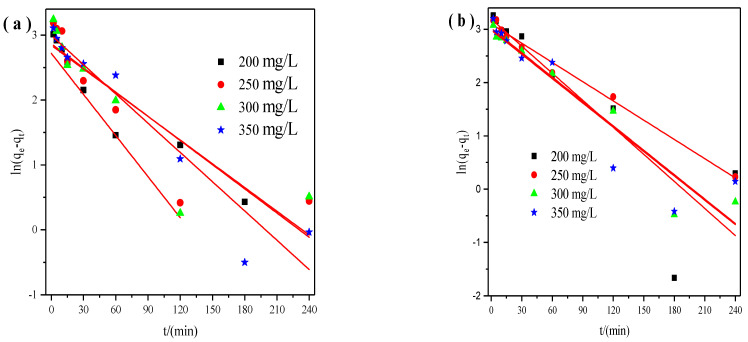
Pseudo−first order model of X zeolite for Cu^2+^ (**a**) and Zn^2+^ (**b**).

**Figure 3 materials-14-06525-f003:**
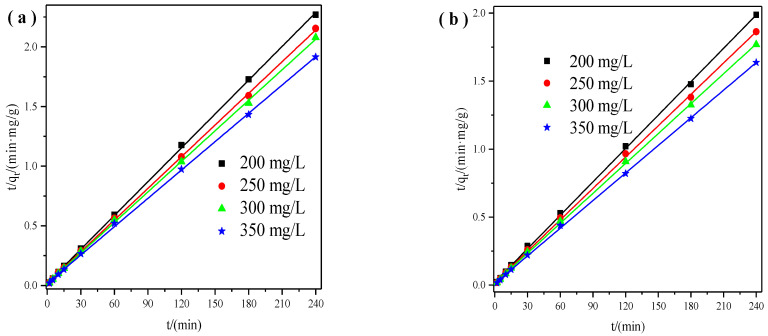
Pseudo−second order model of X zeolite for Cu^2+^ (**a**) and Zn^2+^ (**b**).

**Figure 4 materials-14-06525-f004:**
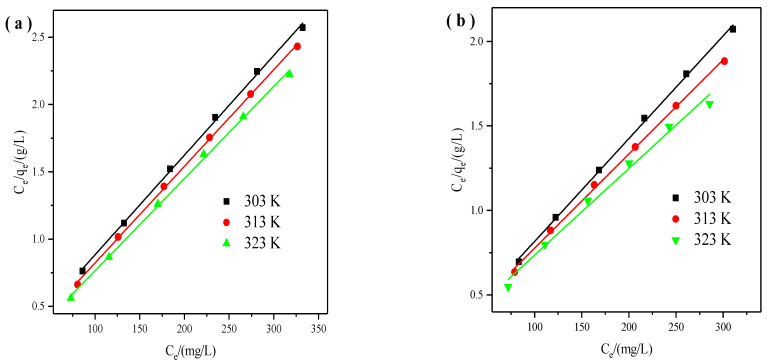
Langmuir isotherms of X zeolite for Cu^2+^ (**a**) and Zn^2+^ (**b**).

**Figure 5 materials-14-06525-f005:**
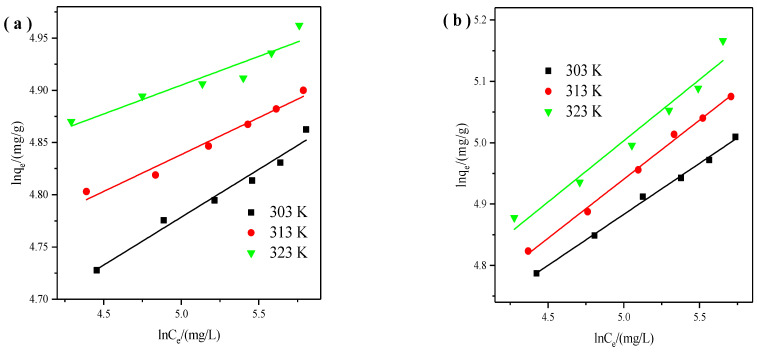
Freundlich isotherms of X zeolite for Cu^2+^ (**a**) and Zn^2+^ (**b**).

**Figure 6 materials-14-06525-f006:**
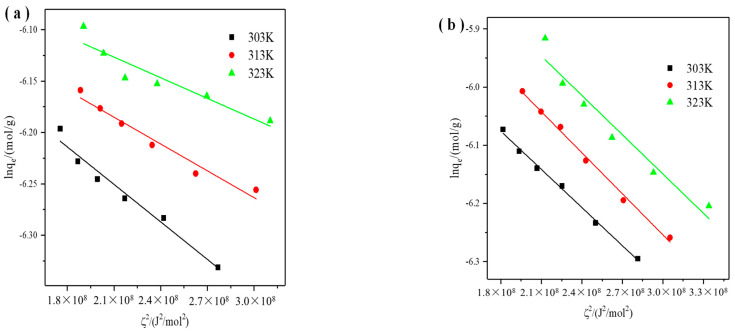
D-R isotherms of X zeolite for Cu^2+^ (**a**) and Zn^2+^ (**b**).

**Figure 7 materials-14-06525-f007:**
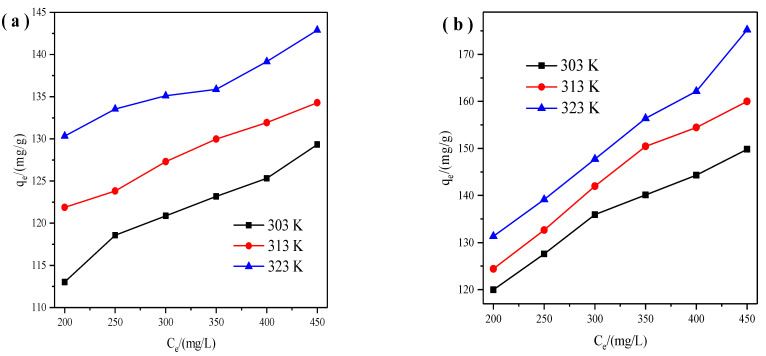
Adsorption isotherms of X zeolite for Cu^2+^ (**a**) and Zn^2+^ (**b**).

**Figure 8 materials-14-06525-f008:**
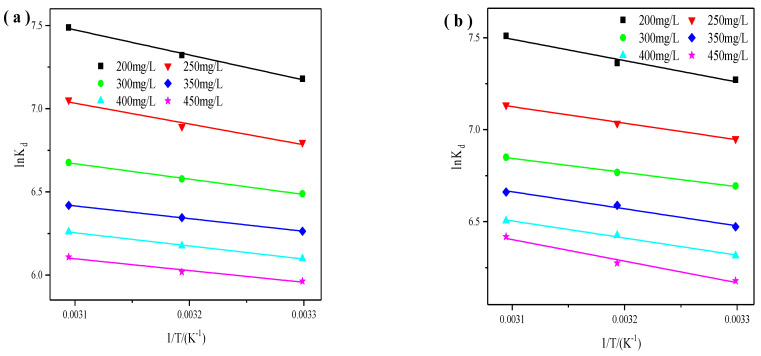
Plot of LnK_d_ versus 1/T at various initial concentrations of Cu^2+^ (**a**) and Zn^2+^ (**b**).

**Figure 9 materials-14-06525-f009:**
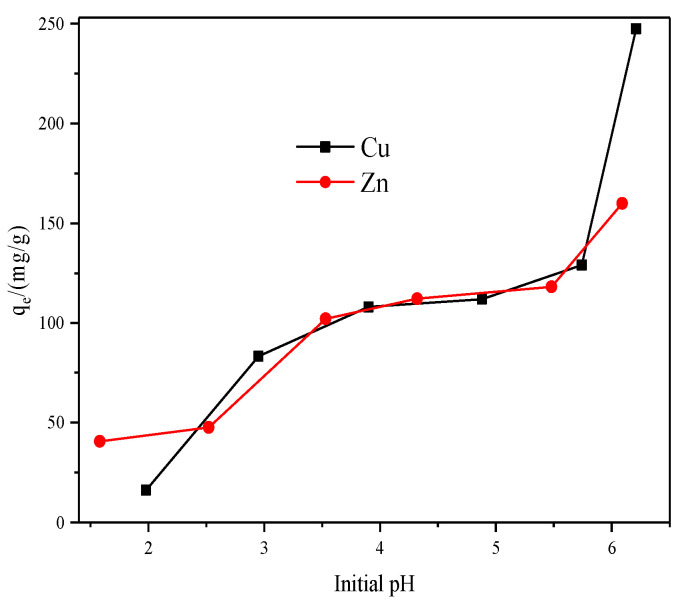
Effect of initial pH on the adsorption capacity of X zeolite for Cu^2+^ and Zn^2+^.

**Figure 10 materials-14-06525-f010:**
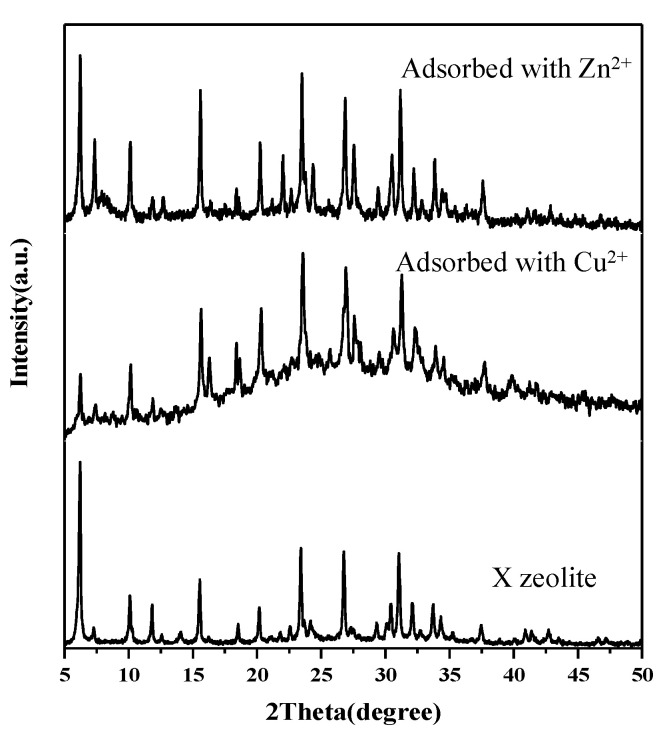
XRD patterns of X zeolite and samples after adsorption with Cu^2+^ and Zn^2+^.

**Figure 11 materials-14-06525-f011:**
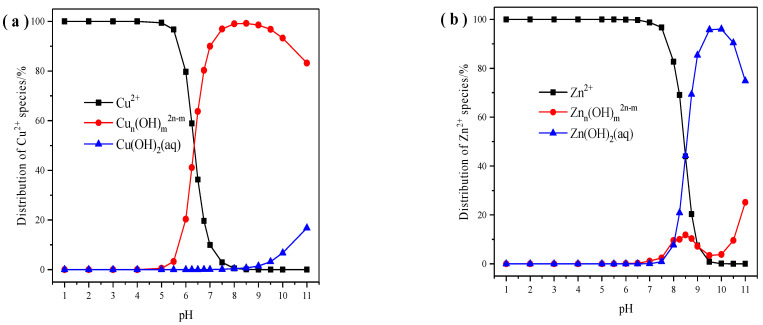
The state distribution of Cu^2+^ (**a**) and Zn^2+^ (**b**) at different pH values.

**Figure 12 materials-14-06525-f012:**
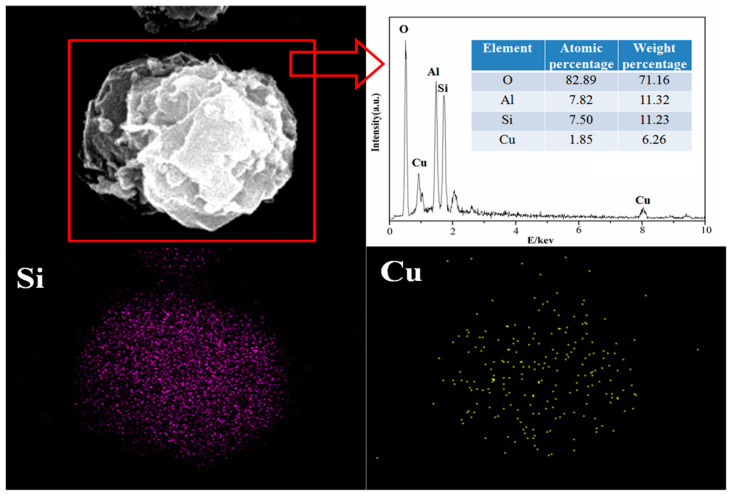
SEM images and EDS analysis of X zeolite adsorbed with Cu^2+^.

**Figure 13 materials-14-06525-f013:**
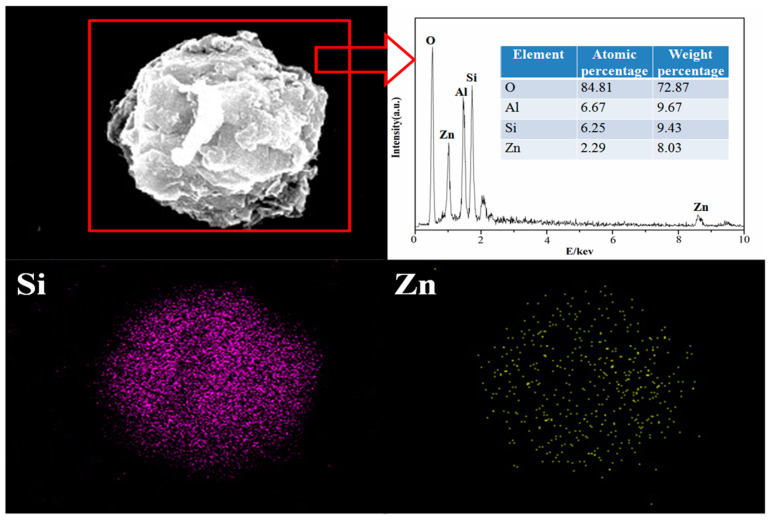
SEM images and EDS analysis of X zeolite adsorbed with Zn^2+^.

**Table 1 materials-14-06525-t001:** Parameters of the pseudo−second order model.

Samples	Cu				Zn			
	k_2_ (g/(mg·min))	h (mg/(g·min))	q_e_ (mg/g)	R^2^	k_2_ (g/(mg·min))	h (mg/(g·min))	q_e_ (mg/g)	R^2^
200	0.00475	53.08	106	0.9997	0.00312	46.51	122	0.9995
250	0.00473	60.39	113	0.9998	0.00330	55.80	130	0.9995
300	0.00471	64.58	117	0.9996	0.00369	68.68	136	0.9997
350	0.00371	59.14	126	0.9996	0.00390	85.09	148	0.9997

**Table 2 materials-14-06525-t002:** Parameters of pseudo−first order model.

Samples	Cu			Zn		
	k_1_ (min^−1^)	q_e_ (mg/g)	R^2^.	k_1_ (min^−1^)	q_e_ (mg/g)	R^2^
200	0.02109	15	0.8617	0.01691	24	0.7519
250	0.01241	18	0.7860	0.01204	22	0.9867
300	0.01217	17	0.7526	0.01526	20	0.9368
350	0.01502	20	0.9160	0.01521	20	0.8668

**Table 3 materials-14-06525-t003:** Parameters of Langmuir and Freundlich isotherms for the adsorption of Cu^2+^.

T/(K)	Langmuir			Freundlich		
	q_m_ (mg/g)	K_L_ (L/mg)	R^2^	K_F_ (mg^1−(1/*n*)^L^1/*n*^g^−1^)	*n*	R^2^
303	135	0.0521	0.9985	75.24	10.92	0.9708
313	139	0.0691	0.9993	88.69	14.15	0.9683
323	146	0.0865	0.9982	102.44	18.14	0.8707

**Table 4 materials-14-06525-t004:** Parameters of Langmuir and Freundlich isotherms for the adsorption of Zn^2+^.

T/(K)	Langmuir			Freundlich		
	q_m_ (mg/g)	K_L_ (L/mg)	R^2^	K_F_ (mg^1−(1/*n*)^L^1/*n*^g^−1^)	*n*	R^2^
303	164	0.0293	0.9985	57.53	6.02	0.9956
313	179	0.0255	0.9984	53.44	5.20	0.9956
323	195	0.0230	0.9881	54.92	5.01	0.9530

**Table 5 materials-14-06525-t005:** Parameters of D-R isotherms for the adsorption of Cu^2+^ and Zn^2+^.

T/(K)	Cu^2+^				Zn^2+^			
	q_m_ (mg/g)	β (mol^2^/J^2^)	E (KJ/mol)	R^2^	q_m_ (mg/g)	β (mol^2^/J^2^)	E (KJ/mol)	R^2^
303	158	1.22 × 10^−9^	20.27	0.9654	221	2.19 × 10^−9^	15.13	0.9947
313	157	0.87 × 10^−9^	23.97	0.9541	253	2.35 × 10^−9^	14.58	0.9933
323	160	0.67 × 10^−9^	27.32	0.8510	273	2.25 × 10^−9^	14.91	0.9373

**Table 6 materials-14-06525-t006:** Comparison of adsorption capacities of various adsorbents for Cu^2+^.

Adsorbent	Synthetic Method	Q_m_ (mg/g)	Refs.
Zeolite from fly ash	Hydrothermal method	64	[36]
A zeolite from coal fly ash	Hydrothermal and impregnation methods	50	[37]
Hematite	Co-precipitation method	84	[38]
α-MnO_2_	Precipitation method	83	[39]
Fe_3_O_4_@APS@AA-co-CA MNPs	Chemical co-precipitation methods	127	[21]
X zeolite	Hydrothermal method	146	The paper

**Table 7 materials-14-06525-t007:** Comparison of adsorption capacities of various adsorbents for Zn^2+^.

Adsorbent	Synthetic Method	Q_m_(mg/g)	Refs.
A zeolite	Hydrothermal and calcination methods	80	[24]
X zeolite	Hydrothermal and calcination methods	68	[24]
A zeolite from coal fly ash	Hydrothermal and impregnation methods	31	[37]
Hydrous manganese dioxide	Precipitation method	57	[40]
Fe_3_O_4_@APS@AA-co-CA MNPs	Chemical co-precipitation methods	43	[21]
X zeolite	Hydrothermal method	195	The paper

**Table 8 materials-14-06525-t008:** Thermodynamic parameters for Cu^2+^ adsorption.

C_0_ (mg/L)	ΔH^0^ (KJ/mol)	ΔS^0^ (J/mol)	ΔG^0^ (KJ/mol)		
			303 K	313 K	323 K
200	12.52	100.92	−18.07	−19.08	−20.08
250	10.38	90.64	−17.09	−17.99	−18.90
300	7.68	79.26	−16.33	−17.13	−17.92
350	6.33	72.97	−15.78	−16.51	−17.24
400	6.58	72.39	−15.36	−16.08	−16.80
450	5.88	68.92	−15.01	−15.69	−16.38

**Table 9 materials-14-06525-t009:** Thermodynamic parameters for Zn^2+^ adsorption.

C_0_ (mg/L)	ΔH^0^ (KJ/mol)	ΔS^0^ (J/mol)	ΔG^0^ (KJ/mol)		
			303 K	313 K	323 K
200	9.66	92.22	−18.29	−19.21	−20.13
250	7.47	82.38	−17.50	−18.32	−19.14
300	6.38	76.69	−16.85	−17.62	−18.39
350	7.70	79.26	−16.32	−17.11	−17.90
400	7.72	78.02	−15.92	−16.70	−17.48
450	9.74	83.43	−15.54	−16.37	−17.21

**Table 10 materials-14-06525-t010:** The composition of X zeolite and samples of X zeolite after adsorption with Cu^2+^ or Zn^2+^.

Samples	Si	Al	Cu	Zn	Na	O	n_(Si/A1)_
X zeolite	21.41	17.07	/	/	14.6	45.33	1.21
Sample adsorbed with Cu^2+^	19.72	16.58	16.14	/	2.40	42.51	1.15
Sample adsorbed with Zn^2+^	18.53	16.01	/	16.69	4.56	42.29	1.12

## Data Availability

The data sets supporting the results of this article are included within the article.

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
