# Peer review of "High Removal Efficiency of Diatomite-Based X Zeolite for Cu2+ and Zn2+"

_materials, 2021, doi:10.3390/ma14216525_

Round 1

Reviewer 1 Report

Manuscript Number: materials-1377447

Dear editor,

In the present paper, Diatomite-based X zeolite was evaluated as an sorbent material for Cu(II) and Zn(II) removal from waters. The results presented by the authors could be of scientific interest. However, the manuscript requires major corrections before publication:

  1. The introduction needs to be improved. Please check the English (e.g. line 3 of introduction, “electroplating” should be electroplating, etc.)
  2. Please add the reference of US EPA (line 5, introduction).
  3. The advantages of sorption process should be highlighted in the introduction. Why is it important to remove these metals from waters? e.g., Sorption may be used for the removal of a big amount of organic and inorganic pollutants (the process is cost-efficient), and can be used in conjunction with several techniques (such as flocculation-coagulation). The authors may use the following reference for supporting these comments:

·         Journal of Environmental Sciences 106 (2021) 105-115; https://doi.org/10.1016/j.jes.2021.01.009

  1. Page 3, line 4: The authors confirm that Diatomite –based X zeolite has high surface area, but not data concerning BET surface have been provided.
  2. Page 3, line 9-10: The authors have provided a relationship between the driving force, concentration gradient and sorption capacity. Please explain it deeper, this is not clear.
  3. Page 3, line 20-21: Please explain in the text, what is the physical meaning of h parameter (Pseudo-second order model)?
  4. Page 8, line 5-6: the authors confirm that adsorption followed a monolayer coverage mechanism. How was it demonstrated? I do not agree. The fitting of the data with a model does not necessarily means that the hypotheses of the model have been satisfied.
  5.  Please arrange the equations 10, 11 and 12
  6.   In the pH studies, the authors have provided the initial pH of the experiments, but what was the final pH? It seems that authors did not anticipate the metal precipitation in the solutions. Please add the final pH in figure 9.
  7. What is the main sorption mechanism involved in the process? Is it sorption or ion-exchange? Please add the discussion in text.
  8. How can be regenerated the sorbent?
  9. Why the proposed sorbent should be used against the commercial ones existing in the market.

Author Response

The authors thank the reviewers for their most helpful and significant contribution. Their comments and suggestions have been included in the revised manuscript.  Please see the revised manuscript.

Reviewer #1: 

In the present paper, Diatomite-based X zeolite was evaluated as an sorbent material for Cu(II) and Zn(II) removal from waters. The results presented by the authors could be of scientific interest. However, the manuscript requires major corrections before publication:

  1. The introduction needs to be improved. Please check the English (e.g. line 3 of introduction, “electroplating” should be electroplating, etc.)

Answer: Thanks for the reviewer’s good advice. We have checked the English in the introduction. Please check the revised version.

  1. Please add the reference of US EPA (line 5, introduction).

Answer: Thanks for the reviewer’s suggestion. We have added the reference of US EPA. Please check the revised version.

  1. The advantages of sorption process should be highlighted in the introduction. Why is it important to remove these metals from waters? e.g., Sorption may be used for the removal of a big amount of organic and inorganic pollutants (the process is cost-efficient), and can be used in conjunction with several techniques (such as flocculation-coagulation). The authors may use the following reference for supporting these comments: Journal of Environmental Sciences 106 (2021) 105-115; https://doi.org/10.1016/j.jes.2021.01.009

Answer: Thanks for the reviewer’s good advice. We have used the following reference. Please check the revised version.

  1. Page 3, line 4: The authors confirm that Diatomite –based X zeolite has high surface area, but not data concerning BET surface have been provided.

Answer: Thanks for the reviewer’s suggestion. We have added the BET surface. Please check the revised version.

  1. Page 3, line 9-10: The authors have provided a relationship between the driving force, concentration gradient and sorption capacity. Please explain it deeper, this is not clear.

Answer: Thanks for the reviewer’s good advice. We have provided the relationship between the driving force, concentration gradient and sorption capacity. Please check the revised version.

  1. Page 3, line 20-21: Please explain in the text, what is the physical meaning of h parameter(Pseudo-second order model)?

Answer: Thanks for the reviewer’s suggestion. We have explained the physical meaning of h parameter and its physical meaning the initial sorption rate. Please check the revised version.

  1. Page 8, line 5-6: the authors confirm that adsorption followed a monolayer coverage mechanism. How was it demonstrated? I do not agree. The fitting of the data with a model does not necessarily means that the hypotheses of the model have been satisfied.

Answer: Thanks for the reviewer’s good advice. According to many literatures, the fitting data of models can indicate the adsorption process fit the model. Therefore, we use the fitting data to determine the best model.

  1. Please arrange the equations 10, 11 and 12

Answer: Thanks for the reviewer’s suggestion. We have arranged the equations 10, 11 and 12. Please check the revised version.

  1. In the pH studies, the authors have provided the initial pH of the experiments, but what was the final pH? It seems that authors did not anticipate the metal precipitation in the solutions. Please add the final pH in figure 9.

Answer: Thanks for the reviewer’s good advice. We have added the final pH in the XRD section of the paper. Please check the revised version.

  1. What is the main sorption mechanism involved in the process? Is it sorption or ion-exchange? Please add the discussion in text.

Answer: Thanks for the reviewer’s suggestion. The adsorption process for Cu2+ and Zn2+ is chemical adsorption and ion exchange, and we have discussed in the text. Please check the revised version.

  1. How can be regenerated the sorbent?

Answer: Thanks for the reviewer’s good advice. The adsorption process for Cu2+ and Zn2+ is ion exchange, and it can be regenerated by sodium chloride solution.

  1. Why the proposed sorbent should be used against the commercial ones existing in the market.

Answer: Thanks for the reviewer’s suggestion. we want to compare the adsorption capacities of the prepared X zeolite for Cu2+ and Zn2+, therefore, we listed the adsorbents in other reported literature.

Reviewer 2 Report

After reviewing the manuscript entitled "High Removal Efficiency of Cu (II) and Zn (II) by X zeolite" and with reference number: materials-1377447, I sent the following comments and observations that I consider should be addressed before its publication in this journal materials.

  1. The authors address a very interesting topic, however the introductory section must be updated with more current references on the different methods of wastewater treatment with heavy metals. Some recommended references:
      https://doi.org/10.1016/j.jes.2021.02.014
    https://doi.org/10.1016/j.cej.2021.130210
    https://doi.org/10.3390/ma13183951
  2.  The authors must use some diagram or representation to graphically explain the copper adsorption mechanism.

3. Can the authors measure the surface charge or zeta potential of materials before and after copper adsorption? This would help to complement the discussion of the efficiency and mechanism of adsorption.

4. Could the authors give an expectation on the treatment of water with heavy metals of more than one metal? Could you try real water?

5. The conclusions section should be improved to highlight the potential use of these results with other types of wastewater that contain more metal ions. 

Author Response

The authors thank the reviewers for their most helpful and significant contribution. Their comments and suggestions have been included in the revised manuscript.  Please see the revised manuscript.

Reviewer #2:

After reviewing the manuscript entitled "High Removal Efficiency of Cu (II) and Zn (II) by X zeolite" and with reference number: materials-1377447, I sent the following comments and observations that I consider should be addressed before its publication in this journal materials.

  1. The authors address a very interesting topic, however the introductory section must be updated with more current references on the different methods of wastewater treatment with heavy metals. Some recommended references: 

https://doi.org/10.1016/j.jes.2021.02.014
https://doi.org/10.1016/j.cej.2021.130210
https://doi.org/10.3390/ma13183951

Answer: Thanks for the reviewer’s good advice. We have used the recommended reference in the introduction. Please check the revised version.

  1. The authors must use some diagram or representation to graphically explain the copper adsorption mechanism.

Answer: Thanks for the reviewer’s suggestion. We have used more diagram or representation to graphically explain the copper adsorption mechanism. Please check the revised version.

  1. Can the authors measure the surface charge or zeta potential of materials before and after copper adsorption? This would help to complement the discussion of the efficiency and mechanism of adsorption.

Answer: Thanks for the reviewer’s good advice. The surface charge or zeta potential of materials before and after copper adsorption is important, and the pH is also important for the copper adsorption. Therefore, we used the initial and final the pH to replace the surface charge or zeta potential. Please check the revised version.

  1. Could the authors give an expectation on the treatment of water with heavy metals of more than one metal? Could you try real water?

Answer: Thanks for the reviewer’s suggestion. We have used tried wastewater with heavy metals of more than one metal in our other literatures. And we have updated the manuscript. Please check the revised version.

  1. The conclusions section should be improved to highlight the potential use of these results with other types of wastewater that contain more metal ions.

Answer: Thanks for the reviewer’s good advice. We have highlighted the potential use of these results with other types of wastewater that contain more metal ions in the conclusion section. Please check the revised version.

Round 2

Reviewer 1 Report

Dear Editor

The present manuscript can be now accepted for publication.

I recommend to the authors to check the English (again).

Author Response

The authors thank the reviewers for their most helpful and significant contribution. Their comments and suggestions have been included in the revised manuscript.  Please see the revised manuscript.

Reviewer #1: 

The present manuscript can be now accepted for publication.

I recommend to the authors to check the English (again).

Answer: Thanks for the reviewer’s good advice. We have carefully checked the English again. Please check the revised version.

Reviewer 2 Report

The authors have responded to the requested comments and the manuscript has been improved. 

In the introductory section, I consider that the authors should cite the corresponding references to this paragraph.

"Zeolites are well-known adsorbents as they can easily adsorb and exchange metal ions with positive charges in their framework. The greatest advantage of zeolites over other adsorbents is that they also show ion exchange selectivity. They have a three-dimensional framework constructed by tetrahedral units and then linked by shared oxygen atoms to create cavities and channels of different dimensions to control ion selectivity in zeolite structures. Furthermore, they possess the net negative charges generated from the isomorphic substitution of Si4+ by Al3+ due to the compensation by alkali and alkaline-earth metal ions. In addition, they are highly porous aluminosilicates with different cavity structures that offers large internal and external surface areas, which are beneficial for adsorption"

Best regards

Author Response

The authors thank the reviewers for their most helpful and significant contribution. Their comments and suggestions have been included in the revised manuscript.  Please see the revised manuscript.

Reviewer #2:

The authors have responded to the requested comments and the manuscript has been improved.

In the introductory section, I consider that the authors should cite the corresponding references to this paragraph. "Zeolites are well-known adsorbents as they can easily adsorb and exchange metal ions with positive charges in their framework. The greatest advantage of zeolites over other adsorbents is that they also show ion exchange selectivity. They have a three-dimensional framework constructed by tetrahedral units and then linked by shared oxygen atoms to create cavities and channels of different dimensions to control ion selectivity in zeolite structures. Furthermore, they possess the net negative charges generated from the isomorphic substitution of Si4+ by Al3+ due to the compensation by alkali and alkaline-earth metal ions. In addition, they are highly porous aluminosilicates with different cavity structures that offers large internal and external surface areas, which are beneficial for adsorption"

Answer: Thanks for the reviewer’s good advice. We have added the corresponding references in the paragraph. Please check the revised version.